# The Quality Characteristics Formation and Control of Salted Eggs: A Review

**DOI:** 10.3390/foods11192949

**Published:** 2022-09-21

**Authors:** Xiaoya Li, Shuping Chen, Yao Yao, Na Wu, Mingsheng Xu, Yan Zhao, Yonggang Tu

**Affiliations:** 1Jiangxi Key Laboratory of Natural Products and Functional Food, Jiangxi Agricultural University, Nanchang 330045, China; 2Agricultural Products Processing and Quality Control Engineering Laboratory of Jiangxi, Jiangxi Agricultural University, Nanchang 330045, China; 3Jiangxi Experimental Teaching Demonstration Center of Agricultural Products Storage and Processing Engineering, Jiangxi Agricultural University, Nanchang 330045, China

**Keywords:** salted eggs, quality characteristics, formation mechanism, affecting factors, control

## Abstract

Salted egg, a traditional characteristic processed egg product in China, is popular among consumers at home and abroad. Salted egg quality characteristics formation primarily includes the hydration of egg white, the solidification of egg yolk, the unique color and flavor of salted egg yolk, and the formation of white, fine, and tender egg whites and loose, sandy, and oily egg yolks after pickling and heating. The unique quality characteristics of salted eggs are mostly caused by the infiltration dehydration of salt, the intermolecular interaction of proteins, and the oxidation of lipids. In recent years, to solve the problems of salted eggs having high salinity, long production cycle, and short storage period, the pickling technology for salted egg has been improved and researched, which has played a significant role in promoting the scientific production of salted eggs. This paper summarizes the mechanisms of salted egg quality characteristics formation and factors influencing quality, with a perspective of providing a theoretical basis for the production of high-quality salted eggs.

## 1. Introduction

With the rapid development of the egg industry, egg processing technology has made great progress and the variety of egg products has continued to increase. Egg products in developed countries in the world are mainly deep-processed products, such as liquid eggs and egg powder. Egg products in China are chiefly traditional products, including salted eggs, preserved eggs, eggs preserved in rice wine (*Zaodan*), and marinated eggs, among which salted eggs account for a large proportion. The salted egg yolks are popular among consumers because of their unique quality of “vermillion oil and fluffy” after being pickled with salt and heated, and are widely used in making egg yolk pastry, mayonnaise, ice cream, and biscuits; the salted egg whites can be used in the production of other food products [1].

The traditional processing methods of salted eggs mostly include the straw ash method, the salt mud coating method, and the brine soaking method. The traditional processing methods of salted eggs suffer from long pickling cycle, an overly salty egg white, and problems such as melting yolks, muddying, black circles, reduced oil output, flocculation, and deterioration of egg whites during storage. To shorten the pickling cycle of salted eggs and improve their quality, the researchers used a combination of traditional and new-fashioned pickling technology for salted eggs [2]. To address the problem of salted egg whites being too salty, two methods are generally used to improve it. The first is to change the pickling method during the pickling process, such as using a two-stage pickling method [3]; the second is to use salted egg whites at multi-levels, such as adding salted egg whites directly to food ingredients [4], desalting salted egg whites [5], or extracting lysozyme, ovalbumin, and bioactive peptides from salted egg whites [6]. To improve the preservation effect of salted egg yolks, cold treatment [7], and immersion of salted eggs in the by-products Garcinia Cambogia extract (Gambier waste) for 10 min [8] can extend their storage life. Problems such as black circles in the cooked salted egg yolks during heating or storage may be caused by using unclean eggs for directly pickling. To solve this problem, raw eggs can be pretreated with hydrochloric acid [9], acetic acid, neutral enzyme [10], or ozone [11] without affecting the flavor of salted eggs. Therefore, a comprehensive understanding of the mechanism of salted egg quality formation and the factors influencing salted eggs quality can not only help to solve the problems such as quality deterioration of salted eggs during storage, but also promote the production of high-quality salted eggs. In this paper, the quality formation mechanism of salted eggs and the factors affecting salted eggs are reviewed and prospected, with a focus on providing a theoretical basis for further research on the quality formation mechanism and the industrialized production of salted eggs (Figure 1).

## 2. Quality Formation Mechanism of Salted Eggs

The processing of salted eggs generally involves pickling raw eggs in a slurry composed of brine or table salt, grass ash, and mud. As pickling proceeds, the raw eggs undergo a series of physicochemical changes, eventually forming the unique quality of the salted egg. The quality characteristics of salted eggs are commonly characterized by indicators such as moisture content, salt content, oil yield, color, and texture.

### 2.1. Egg White Hydration

The higher viscosity of fresh egg white is due to the presence of a thick protein layer, which primarily consists of a combination of ovomucin and lysozyme forming the basis of its gel structure, where ovomucin is a fibrous protein that can be intertwined to form the network structure by intermolecular interactions. During storage of fresh eggs, the complex structure of ovomucin-lysozyme dissociates, resulting in a decrease of egg white viscosity [12]. During the pickling process with salt, the aggregation of protein molecules in egg white is weakened and the protein molecular structure gradually unfolds [13]. As the pickling with salt proceeds, water migrates from the yolk to the egg white also intensifies the loss of viscosity. In addition, the water-holding capacity of egg white results in less migration of free water from the egg white to the outside of eggshell, further exacerbating the hydration of the egg white [14] (Figure 2). The proportion of egg whites by weight increases with the duration of pickling with salt [14]. The high concentration of salt dissociates the ovomucin structure, resulting in egg white hydration [15]. Simultaneously, ovomucin includes a significant amount of acidic amino acids, making itself negatively charged, as well as the concentration of salt ions in egg white may impact the charges of the protein, leading to a decrease in the viscosity of egg white [16].

For the formation process of heat-induced protein gel in salted egg white, it was found that after heat treatment of salted egg whites, the ability of salt to alter the electrostatic interaction between proteins, and metal cations could shield the negative charges carried by the proteins, resulting in less repulsive forces between proteins. Simultaneously, metal ions also had a certain effect on hydrogen bonds, they contested with proteins for water, suppressed the interactions between water molecules and hydrophilic groups of proteins, resulting in the gradual conversion of bound water into free water, the enhanced protein–protein interactions promoted the aggregation of thermally induced denatured protein molecules in egg whites [17,18]. However, as the salt content of egg white increases, the structure of thermally induced salted egg white proteins becomes looser and the cross-linking of gel network weakens [19]. Based on this, it can also be indirectly explained that the formation of a rough and uneven gel network of salted egg white is primarily caused by the hydration of egg white during pickling. To further study the thermal gel changes of egg whites after salt treatment, some researchers have conducted relevant studies by constructing an off-shell model. Kaewmanee et al. [20] discovered that egg whites under the action of higher NaCl concentration could promote the thermal aggregation of egg white proteins through the hydrophobic interaction and ion interaction of egg white proteins, forming rough and opaque gels after heating. Another study indicated that sodium chloride-induced heat denaturation of duck egg whites could break the hydrogen bonds in the gel and reduce the ordered secondary structure (β-sheet and α-helix) of the gel, thus reduce the gel hardness of egg whites [21]. On this basis, Nasabi M et al. [22] investigated the effects of salt and non-ionic surfactants with different concentrations and different heat treatment times on the overall thermal denaturation of egg white proteins and discovered that heat treatment resulted in a transformation between protein secondary structures and indicated heat-induced aggregation of protein molecules.

Salt has a great influence on the viscosity, surface hydrophobicity, secondary structure, and microstructure of egg white, and the changes of these properties directly or indirectly cause the hydration of egg whites, while heat treatment leads to protein aggregation and changes in protein secondary structure in salted egg white. At present, most studies have explained the hydration of salted egg white from the macroscopic perspective of whole-egg pickling, and few have investigated the mechanism of egg white hydration and the main contributors to the pickling process from the perspective of constructing an out-of-shell pickling model for different protein fractions in egg whites. Due to the unique thermal gel properties of salted egg white, in the future product development, it could be considered to directly add salted egg white into food raw materials to improve the gel strength and water-holding capacity of food; it can also be added to food instead of salt to avoid the waste of salted egg white.

### 2.2. Gelling of Salted Egg Yolk

Egg yolk is a natural oil-in-water emulsion in which lipids and proteins are assembled in supramolecular structures at different levels. During the pickling process, the native molecules structures of the yolk undergo changes. Simultaneously, with the continuous effects of osmotic pressure, the solidified part of egg yolk gradually prolongs to the inner yolk, the consistency index gradually increases, and the flow activation energy decreases up to the point where the osmotic pressure inside and outside the egg yolk attains equilibrium, finally the egg yolk changes from semi-fluid viscous condition to solid or semi-solid gel condition [23,24]. In the preliminary period of pickling, salt penetration leads to a gradual increase in the soluble protein content of the egg yolk to the formation of a sol, which enhances the liquidity of water, and the low salt concentration during the pickling period facilitates the exposure of sulfydryl groups. As the curing time increases, egg yolk proteins form soluble aggregates due to intermolecular hydrophobic interactions and new disulphide bonds. In the later period of pickling, the yolk gradually dehydrates to form a gel network structure, and the water fluidity gradually diminishes, with the possibility of sulfhydryl-disulfide exchange reactions and the oxidation of sulfhydryl groups to generate new disulfide bonds under the action of higher salt concentration [23], which plays an important role in the gel formation of salted egg yolk. With the extension of pickling time, the soluble aggregates of egg yolk proteins are formed due to intermolecular hydrophobic interactions and new disulfide bonds. These aggregates, in addition to globular, granular proteins and low-density lipoproteins in egg yolks, also play important roles in egg yolk gel formation [25]. Table salt helps to stabilize proteins in the egg yolk and has a certain “shielding” effect on the repulsion between proteins and promotes the random aggregation of proteins to form a gel [26]. Furthermore, as the salt content in the egg yolk increases, the egg yolk emulsion system is destroyed and the zeta potential value decreases, which causes the egg yolk protein to coagulate at a lower zeta potential value [27]. From the analysis of the microstructural and the protein secondary structure changes in egg yolk during the pickling process, the yolk changes from spherical to irregular polyhedrons and more tightly arranged due to the enhanced chemical bonding in the yolk under the effect of high salt, and the yolk oil is released in the gaps of the yolk sphere [28]. Another study has indicated that the elevated intramolecular *β*-sheet structure content of salted egg yolk proteins is likely because high salt content promoted the rearrangement of hydrogen bonds and the hydrophobic interactions between protein molecules, thus leading to the tightly integration of protein–water–lipid, eventually resulting in a highly viscoelastic yolk gel [23]. We speculate that oil exudation during pickling has a facilitating effect on the formation of salt-induced egg yolk gel. Egg yolk protein concentrates with relatively high lipid content form gel networks at low protein concentrations, suggesting that egg yolk lipid molecules are involved in gel formation in some way [29]. In the granular emulsion, the oil droplets act as active filler to fill the particles, which can enhance the gel strength [30].

Since it is difficult to elucidate the gelation mechanism of egg yolk by directly pickling whole eggs, to further investigate the influence of salt on the gelation of egg yolk, some researchers have constructed an out-of-shell simulated salt pickling system to induce yolk gelation behavior for related studies. It was discovered that when the salt concentration was lower than 1.5% (*w*/*w*), the egg yolk exhibited fluid-like behavior; when the salt concentration reached 1.5% (*w*/*w*), the sol-gel transition occurred in the egg yolk; when the salt concentration exceeded 1.5% (*w*/*w*), the egg yolk demonstrated solid-like behavior [31]. Through the related research on the gelation behavior of egg yolk induced by salt inside and outside the shell, it is concluded that the salt content and the rate of osmotic dehydration are the key factors determining the gel network formation of egg yolk during pickling. The egg yolk can be divided into plasma and granules after dilution and centrifugation [31]. Plasma contains 85% low-density lipoproteins (LDLs), and 15% globular glycoproteins (α-, β- and γ-livetin); granules are composed of 70% high-density lipoproteins (HDLs), 12% LDLs, and 16% phosvitins. To facilitate in-depth research on the mechanism of salt-induced egg yolk gel formation and simplify the complex components of egg yolk, egg yolks were centrifuged [32], and gelation behaviors of plasma and granules were induced by NaCl, respectively. It was discovered that for plasma gels, protein aggregation in the early stage of pickling might enhance the protein-fat interaction, resulting in restricted proton mobility associated with lipid molecules. LDLs are disrupted in the later stage of pickling, releasing phospholipids, neutral lipids, and proteins, which interact to form non-spherical aggregates and accelerate gel formation (Figure 3) [32,33]. Additionally, higher ionic concentration in salt can modify protein molecules, hinder the oxidation of sulfhydryl groups, and affect the formation of disulfide bonds. Meanwhile, the hydrophobic interactions that stabilize the aggregates are still formed to a certain extent. The exchange of intermolecular sulfhydryl-disulfide bonds plays an important role in gel formation [34]. For granule gels, during the pickling process, the granular proteins are disrupted to release more protein components, which may be aggregated by intramolecular and intermolecular protein interactions to form a tighter gel structure. The increase of intramolecular *β*-sheet and intermolecular *β*-sheet can improve the hardness and cohesion of granular gel to a certain extent [32,35]. Additionally, HDLs not only play an important role in improving the gel strength of granules, but also play a special role in the formation of egg yolk with higher viscoelasticity after heating [36]. In addition to LDLs and HDLs, the egg yolk also contains globular glycoproteins and phosvitins; the rich variety of proteins provide the material basis for its gelation behavior [37].

We investigated the gelling mechanism of salted egg yolk and monitored the dynamic changes of moisture and lipid migration in egg yolk during pickling. LF-NMR (low field nuclear magnetic resonance) and its imaging technology, as a non-destructive and rapid analytical tool for food quality evaluation, can not only reflect the dynamic changes of distribution and migration of water and lipids in food system, but also visually describe the migration and distribution of hydrogen protons [38]. Previous studies have discovered that LF-NMR technology could be used to monitor the behavior of water in soybean antioxidant peptide powder in different states [39]. Based on this, a study was conducted to apply LF-NMR technology to the qualitative analysis of the moisture and lipid content of salted eggs during the pickling process, and it is speculated that during the pickling process, the egg yolks coagulate into gels that restrict the movement of moisture and lipids [23,40]. The study also provides the technical and theoretical basis for quality control of salted eggs.

To accelerate the pickling of salted egg yolks, the use of pulsating pressure to pickle salted eggs can promote salt penetration and egg yolk dehydration, which in turn accelerates the coagulation of egg yolk [41]. Additionally, the pickling process is accompanied by liquid smoke, which can also promote yolk solidification [42].

Overall, the penetration of salt changes the physical and chemical properties and microstructure of egg yolks. With the pickling process, the moisture content of egg yolk decreases and the viscosity and hardness increases, which promotes the formation of disulfide bonds and increases intermolecular or intramolecular interactions, thereby facilitating the formation of egg yolk gels. Salt content, permeation rate, and oil exudation are the primary factors affecting the yolk gel formation. Presently, most studies have primarily focused on the induction of egg yolk gel formation under different conditions and on the causes of yolk solidification during the pickling process through index measurement, while the dynamic process of egg yolk solidification and the methods of accelerating egg yolk solidification have been less investigated. In the future, it is necessary to further explore the dynamic process of egg yolk solidification, and based on this theory, some additives or processing methods should be explored to accelerate the solidification process of salted egg yolks and shorten the pickling cycle of salted egg yolks.

### 2.3. Color Formation of Salted Egg Yolk

The color of yolks is an important sensory index to measure the quality of eggs, and the primary factor affecting the coloration of fresh egg yolks is the content of lutein [43]. With the continuous improvement of egg consumption level, the egg yolk color has directly affected the demand of the poultry egg consumption market. In order to meet different consumer preferences for the color of egg products, the addition of natural pigments to the poultry diet to color egg yolk has gradually become a research boom. But the reduced intake of pigments such as lutein and carotenoids due to the different quality of feed sources has led to a gradual lightening of yolk color, which has become an important problem for the development of the egg industry [44]. In view of this, it has been discovered that the addition of plant carotenoids to the diet of laying hens could increase the feed intake and total carotenoid intake of laying hens and improve the coloring effect of egg yolks [45].

Most consumers prefer the natural orange color of salted egg yolks. Based on this, the color change of egg yolks and its formation mechanism during the pickling process has become a research hotspot. A current study has found that during the pickling process, the orange transformation and hardening of egg yolk gradually developed towards the inner yolk. The oxidation of fat and protein in the yolk can lead to the formation of pyrrole pigments, and the continuous accumulation of pyrrole pigments increases the concentration of pigments and leads to the decrease in a* value of yolk [46,47]. It is speculated that the non-enzymatic browning of phosphatidyl ethanolamine and lysine in the egg yolk can also lead to the decrease of a* value [46,47]. The decrease in a* value of yolk signifies that the yolk gradually turns darker from orange. Three color components: L* (luminosity, 0–100 means from black to white), a* (red-green, −a* represents greenness, +a* represents redness) and b* (yellow-blue, −b* represents blueness, +b* represents yellowness) are provided values of three different color by the color meter system [48]. Dang et al. compared the color of salted duck egg yolks and salted chicken egg yolks and found that the L* and b* values both inner and outer of the duck egg yolk gradually decreased; after this time, L* and b* values increased [49]. It was speculated that the color change of salted duck egg yolk might be resulted from the increase in pigment concentration because of the dehydration of egg yolk in the pickling process. Salted chicken egg yolks also darkened during the pickling but did not produce the desired orange color. To further study the color formation mechanism of salted egg yolks, Yang et al. found that with the increase of NaCl concentration, the L* value of plasma in egg yolk increased, and the color of the egg yolk plasma became darker and gradually turned into red-yellow [50]. It may be because pickling with salt could leach egg yolk lipids and increase the dissolved number of fat-soluble carotenoids. NaCl dehydration also caused the egg yolk pigments to become more concentrated and darker. As for the color change of salted egg white during pickling, only a few studies have highlighted that the color change of salted egg white may be caused by lipids seeping into the salted egg white as the yolk matures, resulting in a dramatic shift in both color and flavor of salted egg white [51].

To investigate the effect of different processing methods on the color of salted egg yolk, Singh and Ramaswamy found that egg yolk color change represented by total chromatic aberration (δ E) increased significantly with increasing pressure and processing time under pressure-assisted pickling [52]. After ultrasonic-assisted cooking, the a* and b* values of salted egg yolks increase significantly, in contrast to a remarkable decrease in L* values. The increase in a* and b* values may be attributed to the release of lutein and zeaxanthin encapsulated in lipids, and the ultrasonic treatment promotes the formation of Maillard products. The decrease in L* value is possibly due to ultrasonic intensification of water loss and oil exudation from yolk [53,54].

Commercially available salted eggs are commonly obtained by vacuum-packing and then autoclaving; during the storage or shelf life of salted eggs, the contact layer between the salted egg yolk and the egg white tends to form grey rings, that is, the contact layer gradually changes from orange to grey-green or black. Boiled eggs also produce a grey-green substance when heated, which is similar to the dark circles of salted egg yolks. When the pH value of fresh egg yolk is close to alkaline, the more grey-green substances are produced after cooked at high temperature [55]. Phosvitin has a very strong binding capacity to iron; after the fresh egg is heated, the phosvitin in the outer yolk captures the iron ions bound to the phosvitin in the center of the yolk, therefore, it combines with more iron ions and accumulates in the contact layer between egg yolk and egg white, and acts with H_2_S produced by egg white to form a grey-green substance of phosvitin-sulfur-iron [56]. To determine the formation cause of grey rings in salted egg yolks, Li et al. discovered that the formation of grey rings in vacuum-cooked salted egg yolks were caused by the combined action of sulfur ions, iron ions, and proteins [57]. Adding 0.9% EDTA and 0.8% citric acid to the pickling solution of salted eggs can effectively inhibit the formation of grey rings in salted egg yolks [58]. However, because of the restrictions on the use of food additives, they are not widely used. Based on this, it is necessary to explore metal chelators that meet the standards for the use of food additives to prevent the formation of grey rings in salted egg yolks in the future (Figure 4).

Generally, the color formation of salted egg yolks is closely related to the dehydration of egg yolk, the release of yolk lipid pigments in the yolk, and the oxidation of lipid and protein during the picking process with salt. In preliminary experiments, it was discovered that the pickled salted eggs were treated by three cooking methods, namely steaming, baking, and autoclaving, and then a comparative sensory analysis of the salted eggs revealed that there was quite a difference in the color of the salted egg yolks. The deepening of the color of the salted egg yolks was presumed to be because autoclaving promoted the leaching of oil from the yolk. It is speculated that the oil exudation from the salted egg yolk during the pickling process of salted eggs had a great impact on the color of salted egg yolks. The color of raw egg yolks also affects the color of salted egg yolks. Presently, there are limited studies on the color formation mechanism of salted egg yolks; only a few studies are limited to the characterization of the color of salted egg yolks through chroma measurement or sensory evaluation, and then speculate the reasons for the color change of salted egg yolks, and there is no study to verify and clarify the reasons for the color change of salted egg yolks. In the future, it is necessary to dynamically monitor the color change of salted egg yolks and systematically study the causes of color formation in combination with different model systems (in-shell and out-of-shell pickling systems and salt-induced single fraction of egg yolk), to determine the color formation mechanism of salted egg yolks. Additionally, the color of egg whites also changes during the pickling process, so it is also necessary to carry out relevant research on the color formation mechanism of salted egg whites in the future.

### 2.4. Flavor Formation of Salted Egg Yolk

Flavor substances in food are composed of a variety of compounds that can improve the taste and impart characteristic flavor to food, including non-volatile flavor substances and volatile odor substances, and are produced mainly through biosynthesis, enzymatic reaction, oxidation, and heat decomposition [59]. The flavor research on poultry eggs and their products primarily focuses on volatile flavor substances. Volatile flavor substances are generally pyrolytic and interact easily with other compounds and are therefore highly susceptible to processing conditions. At present, the primary methods used in volatile flavor research include steam distillation, organic solvent extraction, distillation extraction, supercritical carbon dioxide extraction, and molecular distillation [60].

Current research methods for volatile flavor substances in eggs are generally based on the enrichment of volatile flavor substances by methods such as SPME (solid phase microextraction), combined with GC-MS (gas chromatography tandem mass spectrometry) to separate and identify their substance composition. During the processing and storage of eggs, amino acids, reducing sugars, and lipids in eggs undergo a series of reactions such as Maillard and lipid oxidation, and these reaction products can produce a wide range of flavor compounds [59]. Flavor substances in eggs are primarily induced by heating. To investigate the volatile flavor components that contribute to the unique flavor of egg products, Paraskevopoulou et al. found significant differences in the volatile flavor of fresh eggs before and after heating [61]. A study [62] examined the flavor compounds in whole egg, egg yolk, and egg white, respectively. Through comparative analysis, it was discovered that the primary flavor substances in cooked egg yolk were aldehydes and pyrazines, while the main flavor substances in cooked egg white were ketones and pyrazines, and the main flavor substances in the cooked whole egg were alkylbenzenes, nitriles, and ketones. Volatile aldehydes, conversely, were rare in whole eggs and egg whites. Additionally, a study [63] indicated that pyrimidines, pyrazines, and thiazoles were found in heated egg whites, and heterocyclic compounds contribute significantly to the flavor of eggs. In order to delve into the causes of off-flavor in heated egg yolks, it has been discovered that heating drove the leakage of lipids from low-density lipoprotein particles, and lipid oxidation products participated in the formation of off-flavor volatiles in heated egg yolk [64].

Salted egg yolk is the primary source of the special flavor of salted egg, and research on the flavor substances of salted egg yolk has gradually become a hot topic in recent years. To study the flavor components and formation mechanism of salted egg yolk and to promote the research and development of salted egg yolk flavor products, Zhou detected a total of 60 volatile flavor compounds, including 21 alcohols, 11 alkanes, 2 aldehydes, 4 ketones, 7 aromatic compounds, 9 esters, and 5 other substances (1 furan, 1 thiazole, 2 pyrimidines, and 1 dihydroindene) in cooked chicken egg yolk, duck egg yolk, salted chicken egg yolk, and salted duck egg yolk [60]. It was discovered that the flavor compounds in different egg yolks were obviously different, with esters, pyrimidines, and thiazoles likely to be the main flavor compounds responsible for the formation of the distinctive flavor of salted egg yolks, in addition to the unique flavor of salted egg yolk being related to the percentage of each substance in all flavor substances. The study [65] revealed that the flavor of salted yolk positively correlated with the pickling process, it is likely because the flavor of the yolk gradually transforms into that of the salted egg yolk as the pickling proceeds. The unique flavor of salted egg yolk might be the result of the synergistic reaction of fat oxidation and Strecker degradation. In order to rapidly acquire salted egg yolk flavoring base, it was discovered that the use of enzymatic reaction and microwave synergy could induce fresh egg yolk to produce salted egg yolk flavor, the formation of salted egg yolk flavor might be caused by the enzymatic reaction to disintegrate the egg yolk gel network, which was conducive to the migration of water and lipids, and the microwave treatment further accelerated the fat oxidation [66]. It is speculated that the synergistic effect of the two produces volatile compounds such as hexanal, heptanal, benzaldehyde, and 2-amyl furan compounds are presumed to be associated with the flavor of salted egg yolk (Figure 5). For the formation of salted egg white flavor during the pickling process, only a few studies have found that the volatile flavor substances in salted egg whites might originate from the lipid exudation from salted egg yolks, and from the oxidative degradation of free amino acids. Ketones and esters are the main flavor compounds in the salted egg white during pickling process, followed by acids, aldehydes, and alcohols [51].

In summary, the flavor compounds of salted egg yolk may be produced through a combination of Strecker degradation and lipid oxidation during yolk pickling, with pyrimidines, thiazoles, and esters being the flavor compounds specific to salted egg yolks. Salted egg yolk flavor can be induced from fresh egg yolk using an enzymatic reaction in synergy with microwaves. However, most of the current studies focus on what flavor substances are generated during the pickling process of salted egg yolk, and there is no in-depth study on the specific process of salted egg yolk flavor formation. Therefore, further research is needed on the formation process of salted egg yolk flavor substances and methods to induce the formation of salted egg yolk flavor. In addition, the current research primarily focuses on the source, formation rules, and effective methods to reduce the fishy smell of egg yolk after heat treatment. However, there are few studies on the formation mechanism of a fishy smell after the pickling of egg yolk. 

### 2.5. The Formation of Loose, Sand, and Oil Qualities in Salted Egg Yolk

The loose, sandy, and oily characteristic of salted egg yolks is an essential criterion for assessing the quality of salted eggs. As the pickling time prolongs, the dehydration of egg yolk is increased, which results in the enhancement of the connection between egg yolk particles, the yolk particles change from spherical to irregular polygonal, the gap between egg yolk proteins is reduced and the structure is more compact [28]. Cheng et al. discovered that lipids and proteins in salted egg yolks changed from uniformly distributed continuous phases to dispersed granular structures of different sizes [67]. The particle size becomes progressively smaller as pickling proceeds and the number of particles is increased; presumably, the smaller yolk particles may be related to the destruction of lipoproteins [68]. Yolk granules consist mainly of HDL-phosvitin complexes linked by calcium-phosphorus bridge [69]. The structure of natural yolk granules cannot prevent protein denaturation but can avoid the aggregation of LDLs and α-HDLs in different granules [69]. As the pickling time extends, the increase in salt content in the yolk leads to the displacement of Ca^2+^ in the calcium-phosphorus bridge that maintains the structure of the yolk particles by Na^+^, which leads to the destruction of the yolk particles and the dissolution of soluble phosvitin [70]. In conclusion, the shrinkage and aggregation of egg yolk particles may be the primary reason for the formation of the sand texture of salted egg yolks. However, it has been highlighted in the literature that the yolk moisture content would increase if salted eggs had been pickled for too long. It is speculated that the high salt content of the egg white may be due to excessive salt infiltration, and the high salt concentration of salt has a damaging effect on the egg yolk membrane, which in turn causes the water-filling phenomenon in the part near the egg yolk membrane, while the diluted free lipoproteins form a gel after heating [71]. Therefore, when pickling salted eggs, the concentration of the pickling liquid and the pickling time should be controlled to avoid the phenomenon of water-filled yolk as much as possible, to provide a guarantee for the sandiness of yolks after heating.

During the pickling process of salted eggs, sodium ions interact with the negatively charged hydrophobic groups leads to the structure change of LDL in the egg yolk, and breaks the balance between oil and water, resulting in the gradual separation and exudation of emulsified fine lipid droplets originally evenly distributed in the egg yolk, and the eventual aggregation of in the form of large oil droplets due to similar compatibility [72]. In addition, both high-salt treatment and high-salt heat treatment can lead to a decrease in the water content of yolk and an increase in oil exudation [73]. This is also consistent with the results of Ganesan et al., which are that water in egg yolks is expelled during the pickling process, resulting in a higher fat content [74]. The lipids of salted egg yolk consist mainly of triacylglycerol and phospholipid, while the content of diacylglycerol is very low. It has been discovered that the fatty acid composition of salted duck eggs yolks was similar to the fresh yolk, and no free fatty acids were found in the yolk of salted duck eggs, with oleic acid, palmitic acid, and linoleic acid being the most abundant fatty acids in salted duck egg yolks [75]. The poor stability of lipids in salted egg yolks results in poor processing and storage stability of the product. It has been discovered that freeze-dried salted egg yolks could be stored for 90 days, while the stability of salted egg yolk without freezing or spray drying decreased [76]. The researchers found that dietary regulation of laying hens could change the content of saturated and unsaturated fatty acids in egg yolks [77]. Subsequent studies may consider using diet-regulated eggs for salted eggs pickling, and then analyze in comparing the fatty acid content with that of ordinary salted eggs to determine the difference in the nutritional value of salted eggs pickled with different raw eggs.

To explore the processing method that promotes loose, sandy, and oily salted egg yolk, a study [78] revealed that salted eggs pickled for 15 days with a pickling preparation of rice husk ash: salt: water (2:1:1) had the highest content of protein and fat, while salted eggs pickled for 25 days with a salt: water (1:3) pickling solution had the highest general acceptance. Pressure cooking and ultrasonic-assisted cooking are economical and effective processing techniques to improve the quality of salted eggs. Pressure cooking ruptures some irregular particles in the egg yolk, which leads to lipid exudation and a dense oil layer on the surface of the granules. Additionally, pressure cooking can also lead to an increase in saturated fatty acid content, and a slight decrease in unsaturated fatty acid and polyunsaturated fatty acid content [79]. Wang et al. found that ultrasonic-assisted cooking of salted egg yolk led to the collapse of the yolk gel network, which also promoted the free migration of lipids, leading to a sandy texture of salted egg yolk [54]. This is presumed to be indirectly related to the ultrasonic-induced changes in secondary structure, protein thermal denaturation, moisture and lipid distribution, rheology, and microstructure. Table 1 summarizes the primary reasons for the formation of salted egg quality and the methods for inducing/accelerating quality formation.

Generally, with the extension of pickling time, the number of egg yolk particles gradually increases, and the arrangement between particles becomes more and more tight, forming irregular polyhedrons with certain gaps between the particles; and the oil seeps out to fill the gaps, eventually forming the unique loose, sandy, oily texture of salted egg yolks. At present, although studies have highlighted that the causes of the sandy formation of salted egg yolk, the quantitative analysis of these qualities of salted egg yolk has not been clarified. Due to the long cycle of the current traditional pickling method for salted eggs, a model of off-shell separation pickling has been proposed, but the model suffers from the disadvantage that the yolk membrane is prone to rupture, which makes it difficult to realize industrial production. In the future, it is necessary to establish the quality standard of salted egg yolk as soon as possible, and to devote to the establishment of off-shell pickling model to achieve the similar effect as far as possible.

## 3. Factors Affecting the Quality of Salted Eggs

### 3.1. Effect of Pickling Temperature on the Quality of Salted Eggs

As the pickling temperature increased, the lipid content, yolk index, and saponification value of egg yolk increased continuously. Salted eggs of good quality can be obtained at the pickling temperature of 25 °C [80]. When the egg yolks are pickled by separation outside the shell, high temperature has significant effect on the quality of the salted egg yolks although it can accelerate the rate of ion penetration [46]. Based on previous studies, the method of alternating hot and cold pickling salted eggs was adopted. It was revealed that the pickling period of eggs could be shortened to 7–10 d under the environment of alternating high and low-temperature pickling, and the pickling effect was better [81]. The higher pickling temperature accelerates the dissipation of water from the yolk and promotes the formation of yolk gel. Simultaneously, the dissipation of water from the yolk leads to the destruction of the stable emulsification system in the yolk and promotes the exudation of yolk lipids. In summary, higher pickling temperature can accelerate the formation of egg yolk gel and produce salted eggs with more oil [80].

### 3.2. Effect of Salt Concentration on the Quality of Salted Eggs

With the pickling begun, the salt content of salted eggs increases, and the degree of dehydration also gradually increases, resulting in the denser gel network structure of duck egg yolk, the smaller the pores [31]. Under the action of a high concentration of salt, the egg yolk undergoes demulsification, the chemical bond is gradually strengthened, and the gap between egg yolk particles is narrowed, leading to the gradual solidification of the outer of the egg yolk [25]. The dramatic decrease in egg white viscosity in the presence of high salt concentration might be due to the gradual conversion of insoluble ovomucin to soluble ovomucin in the egg white. Simultaneously, sodium ions affect the stretching of the protein molecular structure in the egg white, leading to an increased probability of random aggregation of proteins and the rough texture formed after cooking [28]. With the increase of salt concentration, the oil yield of salted egg yolk increases, and the L* and b* values of yolk also increase, while the values of a* decrease [46]. The salted egg yolks obtained by gradually adding salt are sandier than those obtain by directly configuring a certain concentration of pickling liquid [46]. It is concluded that salt concentration can directly affect the salt penetration rate in the process of pickling, and the penetration rate is an important factor affecting the texture of salted egg yolk sand.

### 3.3. Effect of Pickling Time on the Quality of Salted Eggs

At the extension of pickling time, the salt content of egg white and yolk gradually increase, the moisture content gradually decreases, the moisture content in the egg yolk decreases significantly, the viscosity of the egg white decreases, and the oil yield of the egg yolk increases. The shrinkage of the yolk spheres makes the yolk particles looser, and the oozing oil fill the gaps between the yolk spheres [82]. The study [83] highlighted that, although the oiling effect was very good when the pickling time was too long, too much salt infiltration led to a large difference in salt concentration between the inner and outer of the egg yolk, resulting in serious gelatinization of the egg yolk. Considering this, the pickling time can be shortened by increasing the pickling concentration, reducing the viscosity of egg white, avoiding the porosity blockage of eggshells, or combining the traditional pickling method with the new pickling technology in the subsequent study.

### 3.4. Effect of Pickling Technology on the Quality of Salted Eggs

The traditional pickling method of salted eggs has prevented further qualitative leaps in the industrialization of salted eggs due to the long processing cycle, the unstable quality, and the unrecyclable nature of pickling liquid. In recent years, exploring new salted egg pickling technology, shortening the pickling cycle and improving the stability of salted egg quality has gradually become a research hotspot. There has been research into the application of vacuum decompression pickling technology to the brine soaking method, where the gas in the raw eggs could escape by vacuuming, prompting the pickling liquid to fill the egg with gas and speed up the pickling rate of salted eggs [84]. Another study [85] has indicated that the use of circulating water pickling method could reduce the salt penetration resistance and cause the salt in the pickling solution to enter the eggs in a constant and uniform manner, which also achieved a better pickling effect. The use of ultrasonic technology to pickling salted eggs reduces the viscosity of egg white and the osmotic resistance of salt, while increases the mass transfer rate, thereby shortening the pickling period of salted eggs [16]. It may be because ultrasonic waves can remove the sediments in the pores of eggshell and improve the permeability of egg biofilm. Meanwhile, the cavitation effect of ultrasound changes the internal structure of egg white and yolk protein, reduces the aggregation of egg white protein molecules, and accelerates the infiltration of salt. The use of pulsating pressure technology can also effectively promote the maturation of salted eggs and shorten the pickling time, resulting in salted eggs with oily yolks after 2–3 days of pickling [41]. The reasonable pulsating pressure circulation process causes the concentration and pressure differences inside and outside of the shell, resulting in the mass transfer channels not being blocked due to the flushing action, promoting the infiltration of the pickling solution. Based on traditional salted egg pickling combined with pulsating pressure technology, adding liquor, spices, and citric acid to the pickling liquid can further improve the flavor of salted eggs [86]. In addition, the pickling period of salted eggs can be further shortened by using ultrasonic-pulse pressure pickling technology [87]. To solve the problem of salty egg whites, Zou et al. adopted a two-stage method for pickling salted eggs with low salt, and revealed that when the eggs were disinfected with 30.0–50.0 mg/L saturated chlorine dioxide solution for 3–5 min, then pickled in the first solution of 16.0–20.0% salt concentration for 10 days, then transferred the duck eggs to the second pickling solution with 4.0–6.0% salt, 0.05–0.10% fennel and Sichuan pepper for 20–25 days, it produced high-quality salted duck eggs with low salt content in egg whites and high oil yield in egg yolks [3].

### 3.5. The Influence of Food Additives on the Quality of Salted Eggs

The traditional pickling method usually uses brine as the pickling solution, but after the pickling is completed, it is often unusable because of the complex organic composition, impurities, turbidity, darkening, or odor. To reduce the waste of pickling liquid in salted egg production, the addition of white pepper and black pepper to the pickling liquid allows the pickling liquid to be reused for salted eggs pickling [88,89], and enriches the flavor of salted eggs. The use of carrageenan to regulate the rate of salt penetration at a pickling temperature 28 °C, salt 15%, carrageenan 0.4%, and pickling time 80 days, leading to the salted egg yolk with golden yellow and loose sandy texture, the egg white with moderate salty taste [90]. However, considering the long pickling time, this method is not advisable. In order to achieve high quality salted eggs in a short time, duck eggs are pretreated with 1.0 N HCl for 120 min or pretreated with 5% acetic acid for 30 min before pickling, then pretreated with 0.25% neutral enzyme for 90 min, and finally pickled in aqueous salt solution, which can adjust the permeability of eggshell and effectively shorten the pickling period of salted eggs [9,10]. Tyrosine and phenylalanine are aromatic amino acids, which are precursors of aromatic compounds and are highly correlated with the odor of egg. To reduce the obvious fishy smell of salted eggs, the duck eggs are cleaned and disinfected with 2 g/m^3^ ozone for 60 min before pickling, then pickled with the mud-packing method, which can reduce the fishy smell of salted eggs to a certain extent and achieve a certain disinfection effect [11]. The reason may be that when cleaning duck eggs, ozone molecules enter the egg whites and yolks through biofilms, and the strong oxidative nature of ozone leads to oxidative decomposition of proteins, which causes the protein spatial structure to fold or unfold and amino acid residues such as tyrosine to be exposed to a polar environment.

When cooked salted eggs are cut for consumption, their quality is often compromised by air oxidation. The addition of tea polyphenols and astaxanthin to the marinade not only decreases the black circle and hard core of yolk, but also reduces the oxidation and decomposition degree of yolk lipids [91]. Based on the previous study, an acid pickling agent was obtained by mixing a pH adjusting buffer (0.1 mol/L citric acid and 0.1 mol/L sodium citrate in the ratio of 4.7:15.3) with antioxidant in the ratio of 20:3, which was added to the mud containing more than 25% salt at the ratio of 2.3% to make mud, and the salted eggs were pickled by the mud-packing method. It was discovered that the addition of acidic additives to the mud could not only significantly inhibit the formation rate of the black circle and hardcore of yolk, but also inhibit the decomposition of free amino acids caused by pickling at variable temperature [92]. Adding 0.5% galangal extract to the pickling solution can not only improve the antioxidant activity of salted eggs, but also provide a unique flavor for salted eggs [93]; the addition of *Curcuma phaeocaulis* Valeton concentrate, *Alpinia officinarum* Hance extract, and *Aloe vera* solution to the pickling solution can also increase the antioxidant activity of salted eggs to a certain extent [94,95]. In addition, adding garlic oil to saturated brine to prepare the pickling solution for pickling not only act as a preservative, but also improve the flavor of salted eggs [96] (Figure 6).

As the standard of living improves, people are at increasing risk for hypertension, kidney disease and cardiovascular disease. Low-sodium diets are gradually becoming a general trend because they can increase the basal metabolic rate and avoid elevated blood pressure. Salted eggs are mostly pickled with high concentrations of table salt, resulting in a high sodium content in the finished product. In dry-cured ham, partial substitution of NaCl has a significant effect on salt content and water activity during pickling [97]. Inspired by this, replacing NaCl with KCl in the preparation of salted egg pickling solution increases the antioxidant activity of salted eggs [98]; it also achieves the similar gel strength to that of as salted egg white pickled in table salt alone [99]. Subsequently, Liu et al. found that KCl as a sodium salt substitute accelerated water migration and promoted oil exudation in salted eggs, but it had little effect on the rheological properties and microstructure of salted egg yolks [100]. However, CaCl_2_ as a sodium salt substitute demonstrated the opposite effect on dehydration and oil extraction, and the process of egg white hydration and yolk solidification were delayed. It is speculated that the reason may be because of the formation of carboxyl-bridge bonds by Ca^2+^ during the pickling process of salted eggs, which enhances the gelation properties of egg whites. Part of NaCl can also be replaced by K_2_CO_3_ to promote the penetration of NaCl into eggs and shorten the pickling time [40]. The above studies indicate that K^+^ can be used as an ideal substitute for NaCl, which provide ideas and basis for the subsequent development of low-sodium salted eggs. Table 2 summarizes the main influence, parts, and reasons for different influencing factors on salted eggs.

## 4. Conclusions and Prospects

Salted eggs are well loved by consumers because of their unique qualities. In recent years, the quality formation of salted eggs has progressively attracted the attention of scholars. During the pickling process of salted eggs, the ovomucin structure in the egg white dissociates, and the water in the egg yolk migrates into the egg white owing to the dehydration of the salt, resulting in the gradual hydration of the egg white; the natural egg yolk molecular structure changes, forming disulfide bonds, releasing lipids and granular proteins, which promote yolk gel formation; meanwhile, the release of egg yolk lipid pigment, lipids, and proteins occurs, reducing sugars, and other substances undergo non-enzymatic browning, oxidation, and Strecker degradation reactions, forming the unique color and flavor of salted eggs. After the salted egg yolk is cooked at high temperature, the egg yolk particles gather and the oil gradually oozes out, forming the unique loose, sandy, and oily texture of the salted egg yolk. Although a significant amount of research has been done on salted eggs, the systematic research on the quality formation mechanism and quality control of color, flavor, texture, and oil yield in the processing of salted eggs is far from sufficient. For the not-fully-defined egg hydration mechanism, in the future it may be considered to simplify the pickling system by constructing an out-of-shell pickling system, investigate the effect of major single proteins in egg whites on salt-induced egg white hydration, and filter out the major contributors to egg white hydration during pickling; then researchers would demonstrate the dynamic process of egg white hydration during the pickling of salted eggs in combination with kinetic studies to analyze it. For the study of egg yolk solidification, in addition to the gelatinity of proteins, it is worth analyzing whether the lipid content being about twice as much as protein contributes to the solidification process of salted egg yolk. It is one of the good strategies to investigate the effect of interaction between lipids and proteins on yolk solidification during the salted egg pickling process by using a non-destructive testing technique. Since the color of salted eggs is under the influence of many factors, it has always been a challenge. At present, most studies on the color of salted eggs concentrate on the salted egg yolk, and there are few studies on the color of salted egg white. In the prospective, isotope labeling techniques can be considered to trace the formation process of Maillard products as well as oxidation products to investigate the mechanism of salted egg color formation. With respect to salted egg flavor, it is recommended to further determine the independent or synergistic effects between amino acids, fatty acids, and flavor substances during the salted egg pickling process by combining multiple food flavor detection technologies. For the research on the texture of salted egg yolk after high temperature cooking, future work may focus on the high rate of hard hearts of salted egg yolks, the decrease of oil yield during storage of salted egg, and the grey ring of egg yolk, especially to clarify the mechanism of hard heart formation of salted egg yolks, which will provide the theoretical basis for the control of the texture of salted egg yolk with looseness, sand, and oil.

The pickling methods, pickling time, pickling temperature, salt concentration, and additives directly affect the pickling cycle of salted eggs and the quality of the finished products. The increase of pickling temperature accelerates the infiltration of pickling liquid and the exudation of water in the egg, shortening the pickling cycle of salted eggs. With the growth of salt concentration increases the amount of oil exudation and sandiness in the egg yolk; nevertheless, the degree of hydration of the egg white also enlarges, resulting in a rough texture of the egg white after high temperature cooking. The prolonged pickling time causes a considerable difference in salt concentration inside and outside the egg yolk, which gradually forms a tough gel and loses the unique texture of the salted egg yolk. Adopting ultrasonic assisted pickling technology, pulsating pressure pickling technology, vacuum decompression technology, etc. to pickle salted eggs can widen the mass transfer channel, accelerate salt infiltration, and shorten the pickling cycle of salted eggs. The addition of food additives to the pickling liquid or pickling slurry can regulate eggshell permeability, increase the antioxidant activity of salted eggs, improve the quality of salted eggs, and enrich their flavor. In view of this, increasing the regulation of factors affecting the quality of salted eggs, developing new technologies for salted egg pickling, and searching for pickling auxiliaries for the improvement of salted egg quality are going to become new propositions in the research on salted egg processing.

## Figures and Tables

**Figure 1 foods-11-02949-f001:**
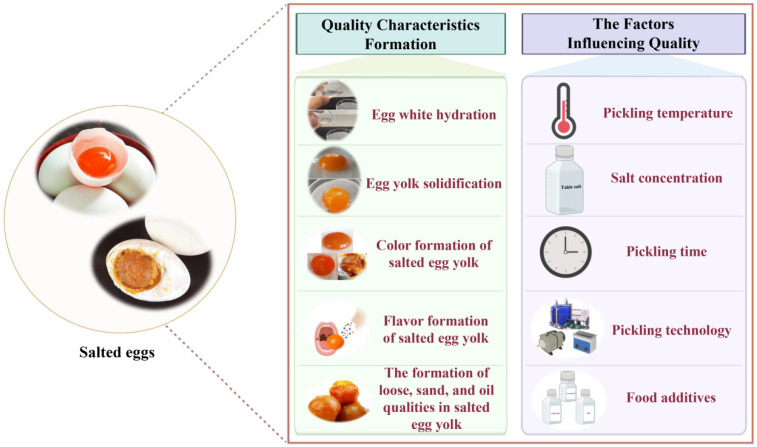
The formation of quality characteristics and the factors affecting the quality of salted eggs.

**Figure 2 foods-11-02949-f002:**
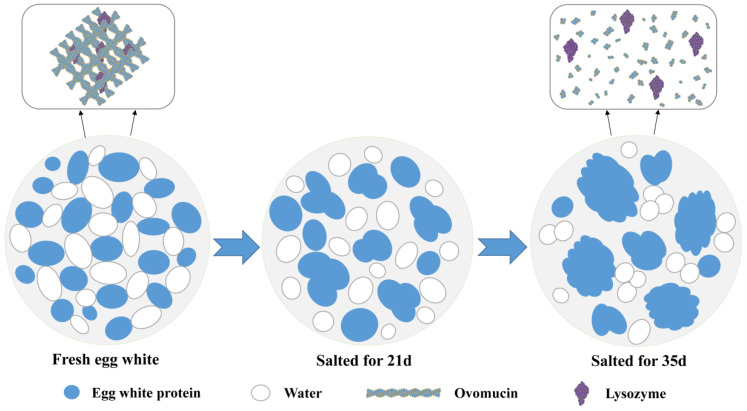
Diagram of hydration mechanism of salted egg white.

**Figure 3 foods-11-02949-f003:**
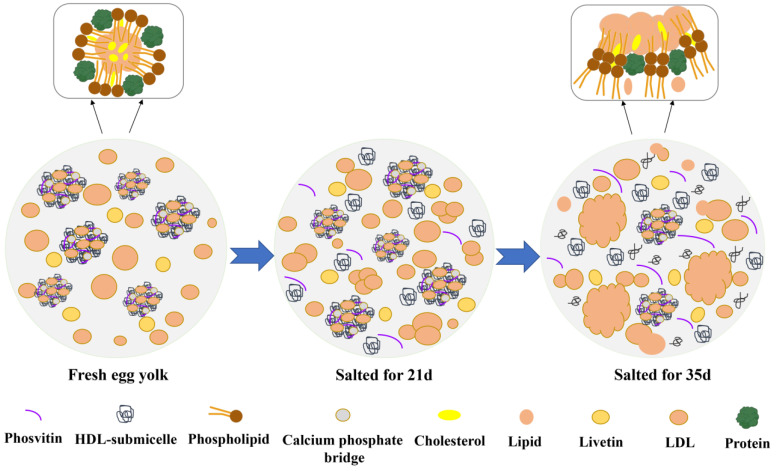
Diagram of gel formation mechanism of salted egg yolk.

**Figure 4 foods-11-02949-f004:**
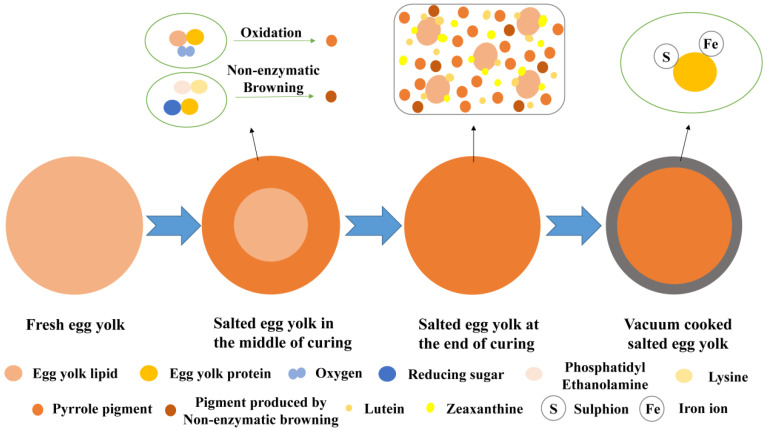
Color formation mechanism of salted egg yolk.

**Figure 5 foods-11-02949-f005:**
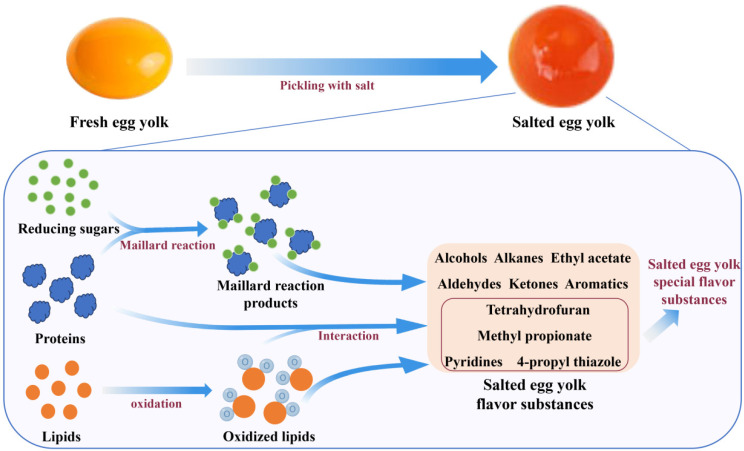
Flavor formation mechanism of salted egg yolk and the flavor substances of salted egg yolk.

**Figure 6 foods-11-02949-f006:**
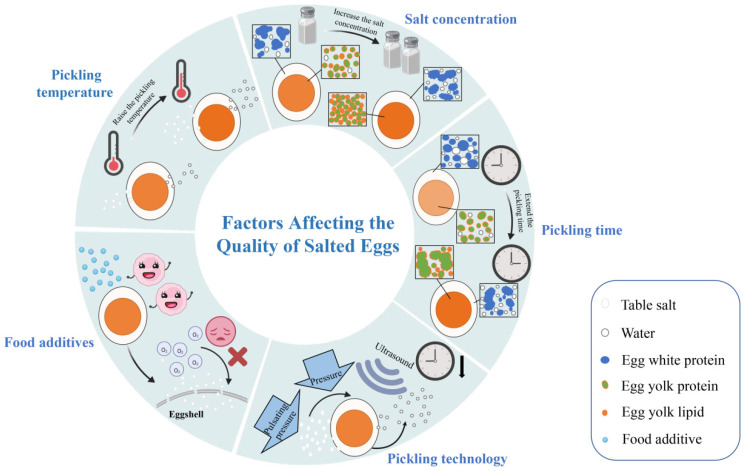
Diagram of the factors affecting the quality of salted eggs.

**Table 1 foods-11-02949-t001:** The quality characteristics formation mechanism of salted eggs.

Quality Characteristics Index of Salted Eggs	The Main Reason for Quality Characteristics Formation	Induction/Acceleration of Quality Characteristics Formation Methods	References
Egg white hydration	A high concentration of salt can dissociate the protein structure and affect the charge of the protein.	Acid treatment of eggshells, ultrasonic treatment	[10,15]
Gelling of salted egg yolk	Salt promotes the molecular interaction of egg yolks and the formation of disulfide bonds, gradually forming the gel network structure.	Pulsating pressure, liquid smoke treatment	[28,41,42]
Color formation of salted egg yolk	Yolk lipids exude, the number of lipid-soluble carotenoids increases, and oxidation of lipids	Ultrasonic treatment, high-pressure vacuum cooking	[47,52,54]
Flavor formation of salted egg yolk	Lipids oxidation and Strecker degradation	Enzymatic hydrolysis and microwave co-processing, high-temperature treatment	[59,60,65]
The formation of loose, sand, and oil qualities in salted egg yolk	Salt breaks down the lipoprotein, causing the yolk particles to become more numerous and the arrangement becomes more and more compact to form irregular polyhedras, and the oil oozes into the particle gaps.	Pressure cooking, ultrasonic-assisted cooking	[26,52,79]

**Table 2 foods-11-02949-t002:** Factors affecting the quality characteristics of salted eggs.

Factors Affecting the Quality Characteristics of Salted Eggs	Main Affected Parts	Influence Reasons	References
Pickling temperature	Egg white, egg yolk	Increasing temperature can accelerate the penetration rate of salt.	[46,80]
Salt concentration	Egg white, egg yolk	The higher the salt concentration, the stronger the chemical bond of the egg yolk protein, which is conducive to the coagulation of the egg yolk.	[20,23]
Pickling time	Egg white, egg yolk	The longer the pickling time, the greater the difference in salt concentration between the inside and outside of the egg yolk, causing the egg yolk to gel.	[14,24,83]
Pickling technology	Egg white, egg yolk	Reduce salt penetration resistance and increase mass transfer rate.	[85,87]
Food additives	Egg shell, egg white, egg yolk	Adjust the permeability of eggshell and penetration rate of salt, improve the antioxidant activity and flavor of salted eggs.	[91,93,96]

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
