# Peer review of "The Quality Characteristics Formation and Control of Salted Eggs: A Review"

_foods, 2022, doi:10.3390/foods11192949_

Round 1

Reviewer 1 Report

This review article has an interesting discussion on the production methods of salted eggs, a typical and widely consumed dish in Asian countries. The text is clear, comprehensive, and relevant to the field.

The review covers aspects of preparation and the technical and methodological aspects that interfere with the final quality of the product. The biochemical review was very interesting and deep. 

In some points, it could be more direct in order not to become boring in reading. 

I appreciate the figures and tables. The figures are easy to interpret and understand and the table properly shows the data. 

References are adequate, but 10% of them are previous 2000. Please consider checking if the information from the listed references is really necessary or if the youngest reference can be used to express the ideas in the text. 

Smith, M.; Back, J.F. Modification of ovalbumin in stored eggs detected by heat denaturation. Nat. 1962, 193, 878-879.

Woodward, S.; Cotterill, O. Texture, and microstructure of heat‐formed egg white gels. J. Food Sci. 1986, 51, 333-339.

Woodward, S.; Cotterill, O. Texture, and microstructure of cooked whole egg yolks and heat‐formed gels of stirred egg 666 yolk. J. Food Sci. 1987, 52, 63-67.

Chi, S.P.; Tseng, K.H. Physicochemical properties of salted pickled yolks from duck and chicken eggs. J. Food Sci. 1998, 63, 717 27-30.

Albright, K.; Gordon, D.; Cotterill, O. Release of iron from phosvitin by heat and food additives. J. Food Sci. 1984, 49, 78-732 81.

Warren, M.; Larick, D.; Ball, J.H. Volatiles and sensory characteristics of cooked egg yolk, white and their combinations. J. 743 Food Sci. 1995, 60, 79-84. https://doi.org/10.1111/j.1365-2621.1995.tb05611.x 744

Umano, K.; Hagi, Y.; Shoji, A.; Shibamoto, T. Volatile compounds formed from cooked whole egg, egg yolk, and egg white. 745 J. Agric. Food Chem. 1990, 38, 461-464. https://doi.org/10.1021/jf00092a028 746

Kato, Y.; Watanabe, K.; Sato, Y. Thermally produced volatile basic components of egg white and ovalbumin. LWT Lebens-747 mitt Wissensch Technol. 1978, 11, 128-130

 Causeret, D.; Matringe, E.; Lorient, D. Ionic strength and pH effects on composition and microstructure of yolk granules. J. 762 Food Sci. 1991, 56, 1532-1536.

 Feeney, R.E.; Weaver, J.M.; Jones, J.R.; Rhodes, M.B. Studies of the kinetics and mechanisms of yolk deterioration in shell 764 eggs. Poultry Sci. 1956, 35, 1061-1066.

 Yang, S.S.; Cotterill, O.J. Physical and functional properties of 10% salted egg yolk in mayonnaise. J. Food Sci. 1989, 54, 210-776 213. 

Reviewer 2 Report

The submitted work entitled ”The quality characteristics formation and control of salted eggs:  A review” presents issues related to the production and quality of salted eggs – as a traditional Chinese product. Due to the quite exotic (regional) nature of the product the work could be not very interesting for other researchers (in the light of international).

The title and abstract reflect the content of the work.

The Authors presented the specificity of the product, emphasized the main advantages and characteristic features of the product like the hydration of egg white, the solidification of egg yolk, the unique colour and flavour of salted egg yolk,  as well as the formation of white, fine and tender egg whites and loose, sandy and oily egg yolks after pickling and heating.

What parameters describe the properties of the product? What values of these parameters affect the quality of salted eggs - for example, maybe the pH? Acidity? Salt content? Maybe the density? What does it mean sandy and oily egg yolks? What parameter describes these features?

Similarly, the lack of technological details - how do the temperature, salt concentration, process time affect on the individual features of the product (can You present it on some a dependence diagram)? What is the value of pressure, ultrasound, etc. in particular technologies?

Details:

Line 35 - the salted egg whites can be used as a “salt substitute” in cooking – substitute means replacement/alternative – salted white egg cannot be a substitute as it contains salt

Lines 171 – 173 - The egg yolk can be divided into plasma and granules after dilution and centrifugation [31], which are composed of 85% low-density lipoproteins (LDLs), and 15% globular glycoproteins (α-, β- and γ- livetin) and 70% high-density lipoproteins (HDLs), 12% LDLs, and 16% phosvitins respectively – is this correct? It gives more than 100%

Line 198 - Low-field NMR – what mean this abbreviation?

Line 202 and 204 - technique or technology - this words are not synonyms

Line 243 - The oxidation of fat and protein in the yolk can lead…. - where does the oxygen in the yolk come from?

Line 245 - leads to the decrease in a* value of yolk - What it means? You should explain what the "a*" means (of course in L*a*b scale of colour), and the same the L* and b*

Line 249 - decreased; After this time - editorial error

Lines 282 – 284 - To determine the formation cause of "black circles" in salted egg yolks, Li et al. found that the formation of black circles in vacuum-cooked salted egg yolks were caused by the combined action of sulfur ions, iron ions, and proteins [55] - "Black circles" on egg yolks are also created  during they boiling (raw eggs). It's usually an effect of to long heat treatment - not salting. It is better use "grey ring" not "black circle"

Line 319 - Volatile flavour substances are generally volatile - it doesn't sound correct

Line 355 - The study [63] found that the flavour of yolk correlated well with the pickling process – what it mean? Positive or negative correlation? What is the relationship?

Line 421 – 424 - The formation of heat-induced emulsified "pork sausage gels" is accompanied by increased protein denaturation, which exposes protein hydrogen bonds and hydrophobic groups, and disulfide bond rearrangement, ultimately leading to escape of free water and fat from the gel network [73] ???

Line 432 - The researchers found that "dietary regulation" - ???

Chapter Factors Affecting the Quality of Salted Eggs – Authors didn’t present specific information. Actually, the information presented in this Chapter are a repetition of Chapter 2.

Conclusions are very laconic. No specific conclusions follow from the presented work.

The selected topic  is interesting, but the work is not properly documented and according to my opinion the paper should have significant changes.

Round 2

Reviewer 1 Report

The main problem detected in the first review was a large number of old references. The suggestion to exchange these references for more recent articles was accepted by the authors and the added references made the review more up-to-date. Thus, I consider that the review has the necessary quality to be accepted and published. 

Reviewer 2 Report

The paper "The quality characteristics formation and control of salted eggs: A review " has been significantly revised and improved.

The Authors revised the manuscript thoroughly according to reviewer's comments and suggestions. The overall scientific quality of the manuscript improved after the revision. Therefore, I suggest accept the paper.